nanotechnology/synthetic chemistry

chlorogenic acid, gold nanoparticles, rapid synthesis, size control, bacteriostatic

**Author for correspondence:**
Sujuan Zhu
e-mail: sjzhu@yzu.edu.cn

# Synthesis of antibacterial gold nanoparticles with different particle sizes using chlorogenic acid

## Sujuan Zhu, Yan Shen, Yongmin Yu and Xuexue Bai

College of Bioscience and Biotechnology, Yangzhou University, Yangzhou, Jiangsu 225009, People's Republic of China

(iD) SZ, 0000-0001-8127-3227

This study proposes a strategy for the rapid and simple synthesis of gold nanoparticles (CGA-AuNPs) with different particle sizes using trisodium citrate (TSC) as the first reducing agent and chlorogenic acid (CGA) as the second reducing agent. And the antibacterial activity of CGA-AuNPs with different particle sizes *in vitro* was checked by measuring the growth curves of *Escherichia coli* (*ATCC 25922*) and *Staphylococcus aureus* (*ATCC 25923*). The CGA-AuNPs obtained by the analysis of transmission electron microscope (TEM) images and ultraviolet–visible (UV–Vis) spectra were mainly spherical, and the average diameters were 18.94 ± 1.81, 30.42 ± 6.32, 37.86 ± 3.80 and 48.72 ± 6.47 nm, respectively. High-resolution transmission electron microscopy (HRTEM) and selected area electron diffraction (SAED) showed that these nanoparticles were polycrystalline gold structures. Both CGA-AuNPs and CGA have excellent antibacterial activity, and CGA-AuNPs with small particle size has a stronger antibacterial effect than the larger one. UV–Vis absorption spectrum data revealed that the synthesized CGA-AuNPs without adding other stabilizing agent were well maintained even after 26 days. This work provides a special idea to regulate the size of CGA-AuNPs with CGA by chemical synthesis, and the potent antibacterial activity of these CGA-AuNPs may be applied in the field of antibacterial in the future.

# 1. Introduction

The vigorous development of nanotechnology has led to the large-scale development and application of nanomaterials. Various nanomaterials with singular functions have been developed and applied in energy, environment, biomedicine and other fields [1–3]. Among them, in the field of biomedicine, nanomaterial-based drugs have brought new hope for the

treatment of difficult diseases such as cancer and AIDS [4–6]. And some nanomaterials have been found to have antibacterial functions [7], such as nano silver [8,9], graphene [10,11], carbon nanotubes [12] and nano-zinc oxide [13,14]. Because these nano-antibacterial materials have the advantages of a broad antibacterial spectrum and no drug resistance. However, these nano-antibacterial materials have certain limitations, including cytotoxicity [15], poor water solubility and poor dispersion [16].

It is worth noting that nano gold also has high antibacterial activity, especially against gram-negative bacteria [17], and it is characterized by good water solubility, high stability, low cytotoxicity and rapid and simple synthesis methods. Synthesis methods of nano gold are mainly divided into physical and chemical methods [18]. The most used ones are the trisodium citrate (TSC) reduction method, the Brust–Schiffrin phase transfer method and the seed crystal growth method [19].

In recent years, scientists have tried to find more green and natural products to synthesize nanomaterials [20–24], because the agents used in these classical synthetic methods are mostly chemical. Chlorogenic acid (CGA) is a kind of natural compound with good antibacterial function, which exists in honeysuckle and other plants. And CGA mainly plays an antibacterial role by destroying the permeability of cell membrane [25,26]. Hwang *et al.* successfully synthesized gold nanoparticles (CGA-AuNPs) using natural compound CGA as a reducing agent [20]. Scaiano *et al.* reported a photochemical synthesis method to regulate the size and shape of nanoparticles [27]. However, there are relatively few studies using the natural compound for the synthesis and regulation of the sizes of gold nanoparticles.

In this experiment, we chose CGA with antibacterial properties as one of the reducing agents for the synthesis of CGA-AuNPs with different particle sizes. The synthesis of different particle sizes of CGA-AuNPs regulated by CGA was characterized by ultraviolet–visible (UV–Vis) spectra, transmission electron microscrope (TEM), high-resolution transmission electron microscopy (HRTEM) and selected area electron diffraction (SAED). The antibacterial properties of CGA-AuNPs were verified by measuring the effect of CGA-AuNPs on the growth curves of *Staphylococcus aureus* (*ATCC 25923*) and *Escherichia coli* (*ATCC 25922*). And the antibacterial effect of different sizes of CGA-AuNPs was viewed using an optical microscope. In addition, the absorbance of CGA-AuNPs with different sizes was measured by an ultraviolet spectrophotometer to evaluate the stability of CGA-AuNPs. The results showed that we have successfully synthesized CGA-AuNPs with different particle sizes and good stability, and CGA-AuNPs with different particle sizes also have the antibacterial effect. Under the same particle size, the antibacterial effect of CGA-AuNPs on *S. aureus* (*ATCC 25923*) was significantly better than *E. coli* (*ATCC 25922*).

# 2. Results and discussion

## 2.1. UV–visible spectra analysis

Under the synthesis condition of 100°C, the concentration of $HAuCl_4 \cdot 3H_2O$ and CGA ($1.18 \times 10^{-6}$ mol $l^{-1}$) in the reaction system was kept unchanged. With the increase of the concentration of TSC in the system, the position of the characteristic absorption peak of CGA-AuNPs was obviously blue-shifted (figure 1), indicating that the particle size decreases with the increasing concentration of the reducing agent TSC. At the same time, the absorbance of CGA-AuNPs also increased, reaching the maximum absorption when $n(Au^{3+}):n$ (TSC) = 1:12. With the increase of the concentration of TSC, we found that the absorption value of CGA-AuNPs no longer increased and the characteristic absorption peak no longer shifted blue, indicating that almost all the gold ions in the system were reduced [28].

Adding $HAuCl_4 \cdot 3H_2O$ and TSC in the same concentration (molar ratios are 1:12) as above at 100°C. The CGA-AuNPs prepared with the increase of CGA concentration gradually shifted to the long-wave direction (figure 2), indicating that the particle size was gradually increasing. The reason for this phenomenon can be explained that the number of gold atoms formed has been stable after the addition of TSC. And then added CGA will be adsorbed on the surface of gold atoms, and neutralize the negative charge on the surface of particles. With the increase of CGA, the electrostatic repulsive force between particles decreases gradually, thus promoting the aggregation of particles, and finally the average particle size increases.

When the molar ratios were 1:0.7, 1:1.05, 1:1.40, it was found that the UV–Vis absorption spectrum of CGA-AuNPs had a distinct absorption peak characteristic of CGA (figure 2*f*) in the wavelength range of 200–350 nm (figure 2), indicating that CGA was adsorbed on the surface of gold nanoparticles.

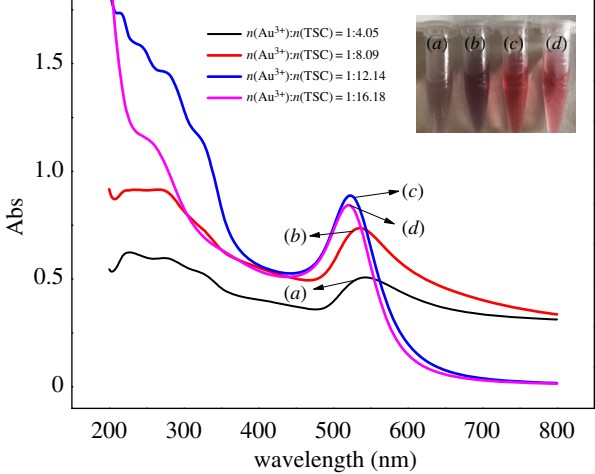

**Figure 1.** UV–Vis absorption spectra of CGA-AuNPs prepared with TSC at different concentrations.

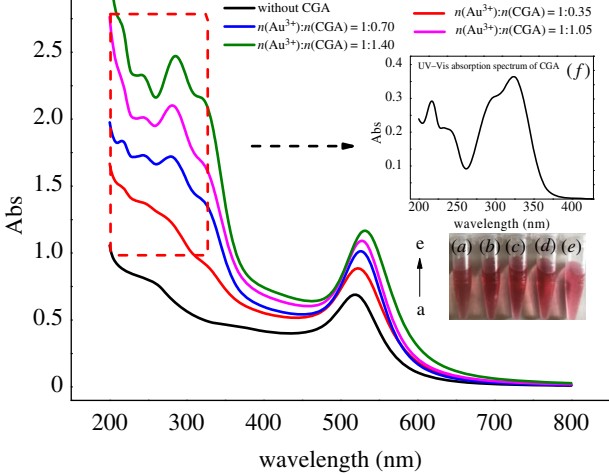

**Figure 2.** UV–Vis absorption spectra of AuNPs without CGA (*a*), CGA-AuNPs were prepared by CGA at different concentrations (*b–e*) and CGA (*f*).

## 2.2. TEM analysis of CGA-AuNPs

The TEM images and size histogram of CGA-AuNPs with different particle sizes are shown in figure 3. It can be seen from the TEM images that the shape of these CGA-AuNPs was mainly spherical and the particle sizes gradually increases.

Figure 3*a* shows the TEM image of AuNPs without CGA synthesis under the same conditions, and the average particle size was $14.45 \pm 1.39$ nm. Figure 3*b–e* shows the TEM images of CGA-AuNPs synthesized with the increase of CGA in the system; the corresponding average particle sizes were $18.94 \pm 1.81$, $30.42 \pm 6.32$, $37.86 \pm 3.80$ and $48.72 \pm 6.47$ nm, respectively. Since the synthesized nanoparticles were not the perfect spherical shape, there will be an inevitable deviation in the statistical particle sizes. And the CGA of different particle sizes (100 particles per condition) were compared by one-way ANOVA. It was found that there was a significant difference ($F = 956.414$, $p < 0.01$) between each group, and the particle size of the nanoparticles has a tendency to become larger as the concentration of CGA increases (figure 3*f*), indicating that CGA can indeed regulate the particle sizes within this range, which was consistent with the UV–Vis absorption spectra of CGA-AuNPs (figure 2).

## 2.3. HRTEM and SAED analysis of CGA-AuNPs

The crystallization degree of five kinds of CGA-AuNPs was observed by HRTEM. As shown in figure 4, the lattice stripes of five kinds of nanoparticles with different particle sizes were clearly visible, and the

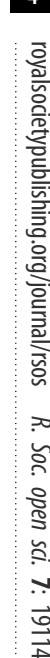

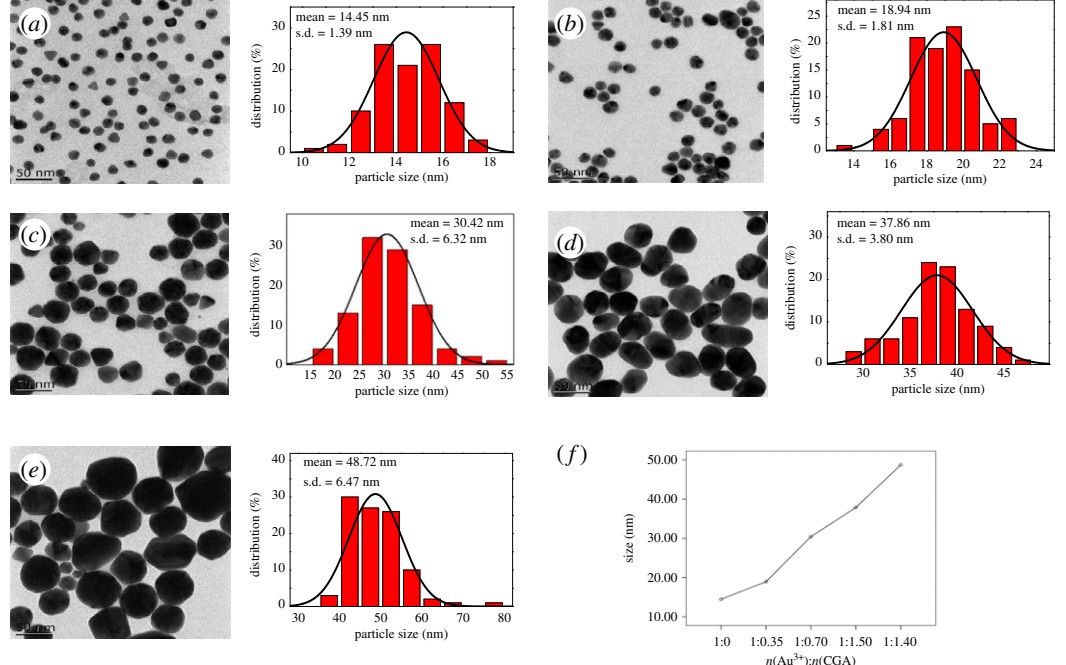

**Figure 3.** TEM images of AuNPs (*a*) and CGA-AuNPs (*b*–*e*) with different sizes. Relevant histogram analysis (right) and mean plots (*f*).

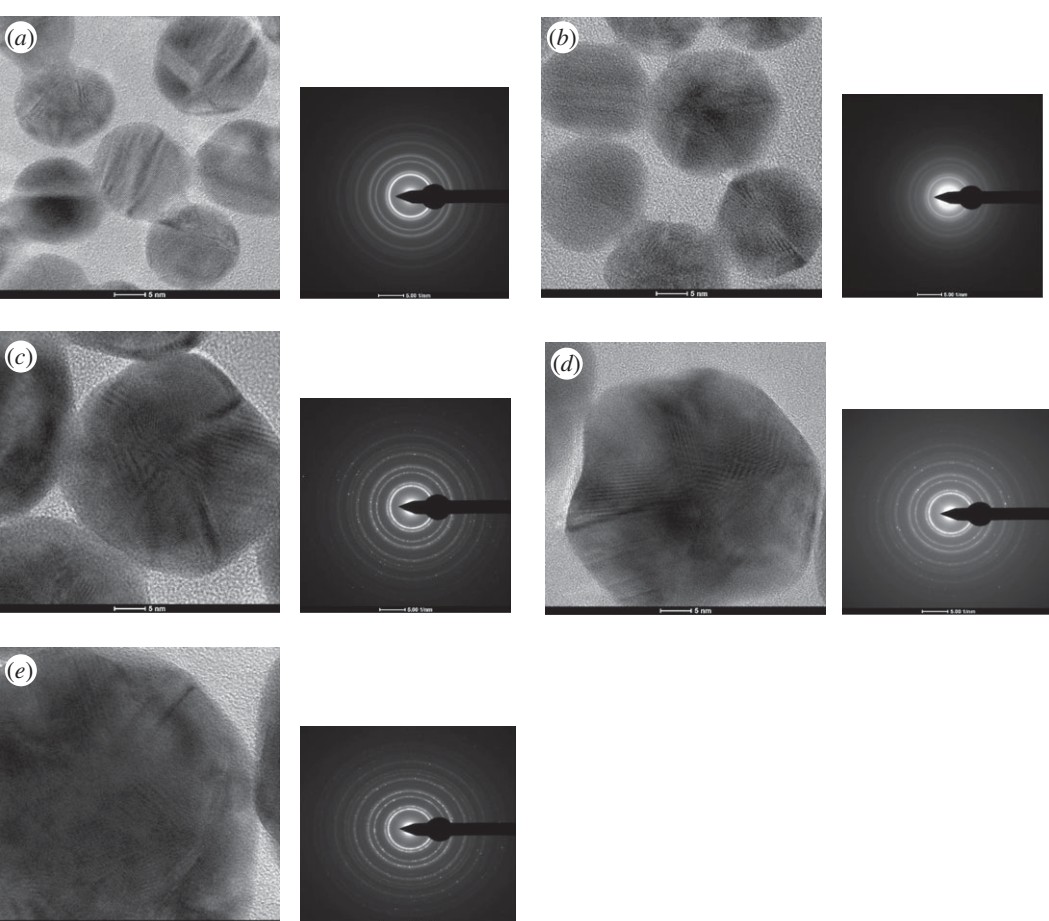

**Figure 4.** HRTEM images of AuNPs (*a*) and CGA-AuNPs from 18.94 ± 1.81 to 48.72 ± 6.47 nm (*b*–*e*), and SAED patterns (right of the HRTEM).

lattice fringes were uniformly distributed. The lattice spacing was 0.24 nm, indicating that all of them were crystal structures. To the right of the HRTEM images in figure 4 are corresponding electron diffraction patterns of these nanoparticles. It was calculated that the diffraction ring in the diagram

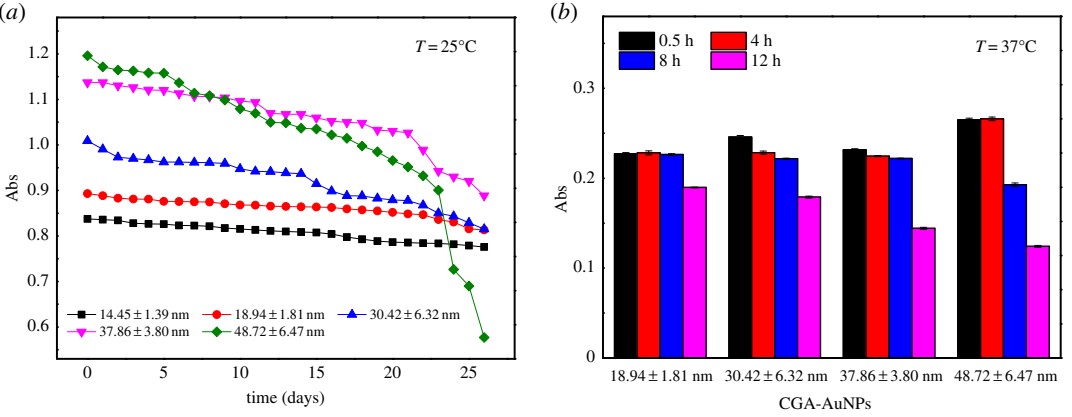

**Figure 5.** Absorbance of CGA-AuNPs colloid with different particle sizes at 25°C over 26 days (*a*), and absorption value of CGA-AuNPs (18.94 ± 1.81 nm, 30.42 ± 6.32 nm, 37.86 ± 3.80 nm and 48.72 ± 6.47 nm) at 520 nm for 12 h at 37°C in a 50% LB medium (*b*).

corresponds to the (311), (220), (200) and (111) crystal planes in the standard PDF card (JCPDS Card-No.PDF 04-0784) of gold elements from inside to outside, respectively. It was indicated that these synthesized nanoparticles were polycrystalline gold structure.

## 2.4. Stability of CGA-AuNPs

In this experiment, the maximum absorptions wavelength of CGA-AuNPs colloids with different particle sizes were measured at 25°C. AuNPs (14.45 ± 1.39 nm) without CGA were used as the control group. As can be seen from figure 5*a*, the absorption value of 18.94 ± 1.81 nm CGA-AuNPs and the control remained unchanged after 26 days. Compared with large-sized CGA-AuNPs, small-sized CGA-AuNPs were more stable in colloid.

The absorption values of CGA-AuNPs with different particle sizes at 520 nm in a 50% LB medium were also measured at 37°C within 12 h, and the results were shown in figure 5*b*. Most of the CGA-AuNPs were stable within 8 h, and the absorption value of CGA-AuNPs of 48.72 ± 6.47 nm decreased slightly, which indicated that CGA-AuNPs had good stability in 50% LB medium at 37°C, and the stability of large particle size was slightly worse, which was consistent with the stability trend shown in figure 5*a*.

## 2.5. Bacteriostatic effect of CGA-AuNPs

The antibacterial activity of nanoparticles is influenced by many factors, in addition to the sizes of their own particles, and their shape, surfactant and type of reducing agent [29]. The effect of CGA-AuNPs prepared by different amounts of CGA on the growth curve of *E. coli* (*ATCC 25922*) and *S. aureus* (*ATCC 25923*) are shown in figures 6 and 7. The different concentrations of CGA were used as the control group. The same CGA concentrations of CGA-AuNPs with different sizes were the experimental group. From figures 6*f* and 7*f*, it can be seen that the antibacterial effects of CGA-AuNPs (18.94 ± 1.81 nm) and CGA are basically the same, and the antibacterial effect was significantly reduced as the particle size become larger. ANOVA results showed that there were significant differences in the antibacterial effect between different particle sizes ($F = 159.345$, $p < 0.01$). This showed that particle size was one of the factors affecting the antibacterial effect. The possible reason is that small-sized CGA-AuNPs have a larger specific surface area and are, therefore, more likely to interact with cells. In addition, the level of bacteriostatic rate was also related to the CGA concentration. The bacteriostatic effect of different CGA concentrations was different ($F = 183.169$, $p < 0.01$). As shown in figure 7*g* and *h*, under the same conditions, the antibacterial effect of CGA-AuNPs on *S. aureus* (*ATCC 25923*) was higher than that on *E. coli* (*ATCC 25922*), which was consistent with the results of the control group. ANOVA showed that there were differences in the antibacterial effect between different bacteria ($F = 110.518$, $p < 0.01$). The reason for this phenomenon may be related to the surface reducing agent CGA. The presence of CGA may alter the interaction between nanoparticles and the two bacteria.

In order to more clearly illustrate the antibacterial activity of CGA-AuNPs with different particle sizes, the morphological changes of *S. aureus* (*ATCC 25923*) and *E. coli* (*ATCC 25922*) treated with

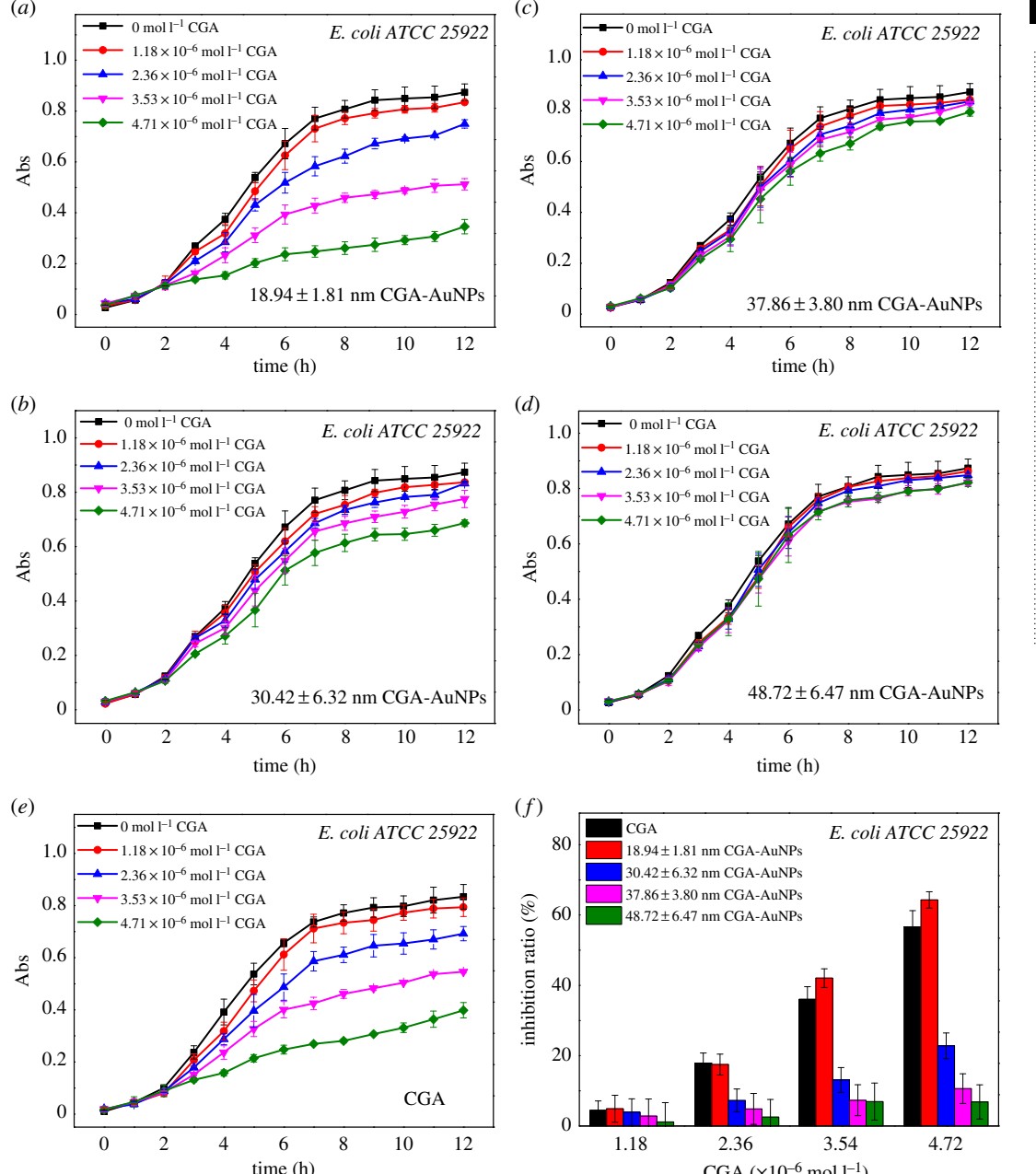

**Figure 6.** Effects of CGA-AuNPs with different particle sizes ($a$–$d$, 18.94 ± 1.81, 30.42 ± 6.32, 37.86 ± 3.80 and 48.72 ± 6.47 nm) and CGA ($e$) on the growth curve of *E. coli* (*ATCC 25922*). ANOVA of the inhibition rate of *E. coli* (*ATCC 25922*) by different particle sizes and concentrations of CGA ($f$).

CGA or CGA-AuNPs were observed by optical microscopy. As shown in figure 8$a$ and $f$, the two bacteria that were not added to CGA-AuNPs and CGA were intact, *S. aureus* (*ATCC 25923*) shown Gram-positive (blue-violet), and *E. coli ATCC 25922* shown negative (red). After treatment with CGA-AuNPs for 8 h, the shape and size of the two bacteria changed, showing swelling, deformity and fragmentation, which may be caused by the accumulation of CGA-AuNPs inside the bacteria and the influx of water. Some of the *S. aureus* (*ATCC 25923*) was dyed red (figure 8$c$–$e$), indicating that the cell membrane was severely damaged. And the two bacteria treated by CGA (figure 8$b$ and $g$) mainly showed swelling, but the morphology was basically intact, and the damage degree of the cell membrane was relatively small. This showed that the mechanism of CGA-AuNPs on bacteria was not exactly the same as CGA.

Bacteriostatic experiments showed that the size of CGA-AuNPs is crucial for enhancing the antibacterial activity, so exploring the use of CGA to synthesizes smaller particle size CGA-AuNPs will be our future work. Meanwhile, CGA-AuNPs not only have good bacteriostasis, stability and

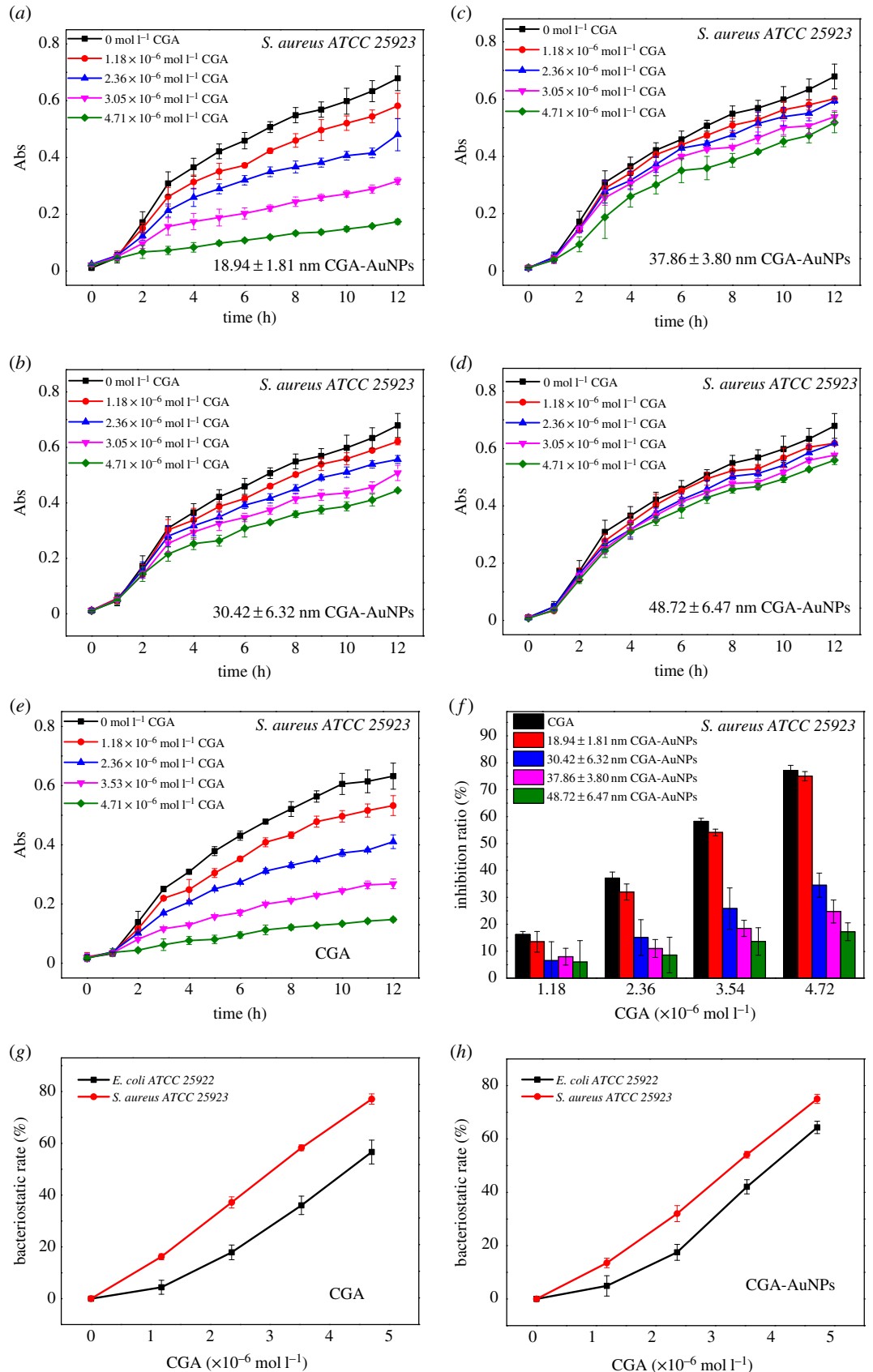

**Figure 7.** Effects of CGA-AuNPs with different particle sizes (*a–d*, 18.94 ± 1.81, 30.42 ± 6.32, 37.86 ± 3.80 and 48.72 ± 6.47 nm) and CGA (*e*) on the growth curve of *S. aureus* (*ATCC 25923*). ANOVA of the inhibition rate of *S. aureus* (*ATCC 25923*) by different particle sizes and concentrations of CGA (*f*). The inhibition of CGA (*g*) and CGA-AuNPs (*h*) on *E. coli* (*ATCC 25922*) and *S. aureus* (*ATCC 25923*).

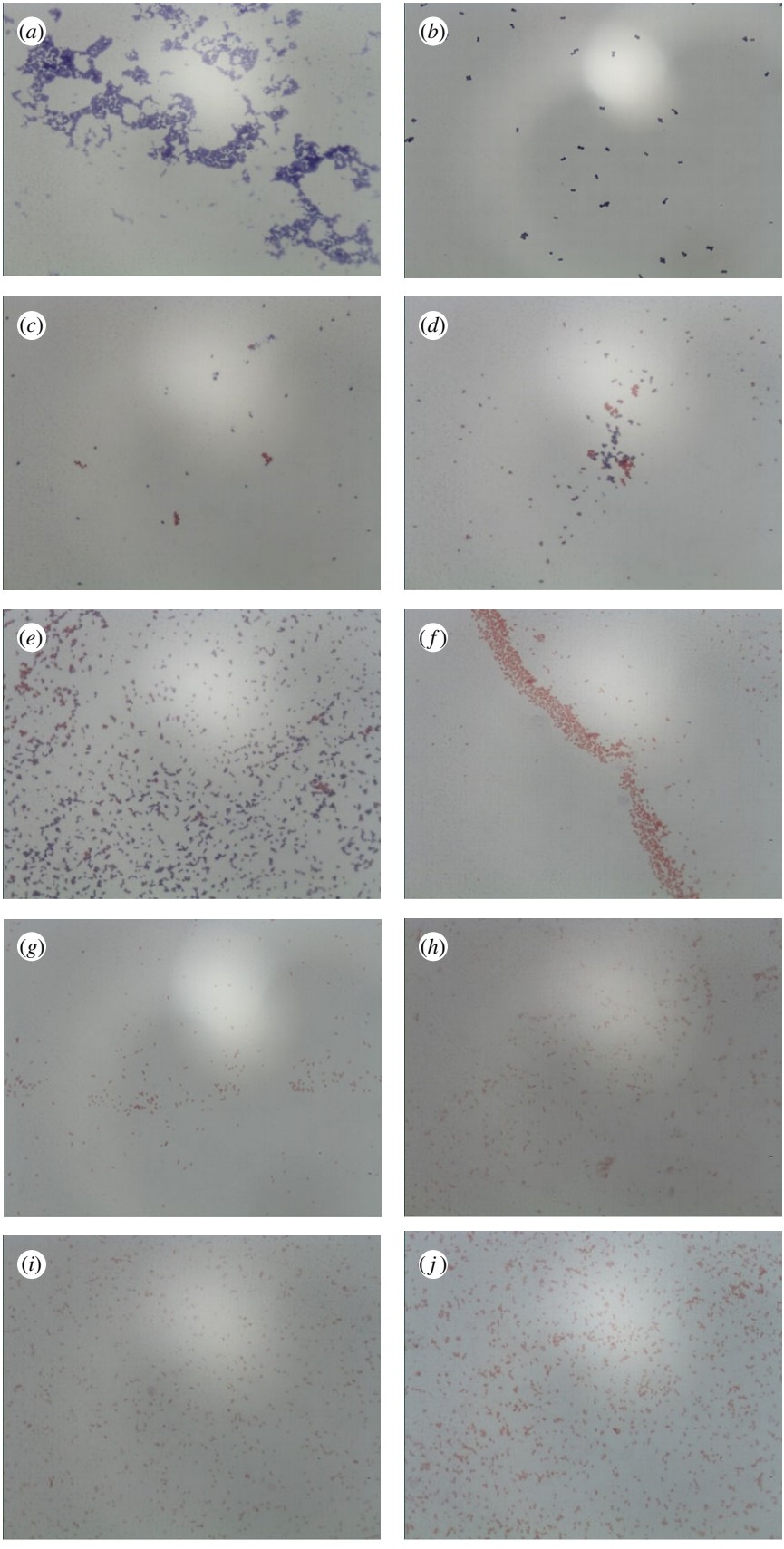

**Figure 8.** Photos of Gram staining of *S. aureus* (*ATCC 25923*) (*a–e*) and *E. coli* (*ATCC 25922*) (*f–j*). (*a,f*): Control test; (*b,g*): test with CGA; (*c,h*): test with 18.94 ± 1.81 nm CGA-AuNPs; (*d,i*): test with 30.42 ± 6.32 nm CGA-AuNPs; (*e,j*): test with 48.72 ± 6.47 nm CGA-AuNPs (×1000 magnification).

biocompatibility but also have the advantages of nano gold, such as small size effect, optical effect, surface effect, low cytotoxicity and its modifiability, which make CGA-AuNPs have infinite potential in many fields waiting for our exploration.

# 3. Experimental section

## 3.1. Materials and instruments

Hydrochloroauric acid trihydrate ($HAuCl_4 \cdot 3H_2O$) were purchased from Shanghai Fine Chemical Materials Research Institute (Shanghai, China). TSC and CGA were purchased from Shenggong Biological Engineering Corporation (Shanghai, China). Strains of *E. coli* (*ATCC 25922*) and *S. aureus* (*ATCC 25923*) and Gram stain kit were purchased from Huankai Microbial Technology Corporation (Guangdong, China). All the chemicals were of analytical grade and were used without further purification. All the solutions were freshly prepared using ultra-pure water.

UV–Vis spectra were recorded on a Cary 5000 (Varian Corporation, USA). TEM images were obtained on a Tecnai 12 model (Philips Corporation, The Netherlands). HRTEM and SAED were obtained on Tecnai G2 F30 S-TWIN (FEI Corporation, USA).

## 3.2. Synthesis of CGA-AuNPs with different particle sizes

For the synthesis of the CGA-AuNPs, a hydrochloroauric acid trihydrate solution (0.01%, in ultra-pure water, 100 ml) was placed into a glass flask and heated to boil on a hot plate (set at 100°C) with stirring. To this solution, a solution of TSC (1%, in ultra-pure water, 3 ml) and CGA were added order and stirred on the hot plate for 12 min. Then, ultra-pure water (dilute to 100 ml) was added when the reaction mixture was naturally cooled. CGA-AuNPs of different particle sizes can be obtained by adding different volumes of CGA (0.25 mM, in ultra-pure water, 0, 2, 4, 6 and 8 ml) according to the procedure described above.

## 3.3. *In vitro* antibacterial activity of CGA-AuNPs with different particle sizes

The antibacterial activity of CGA-AuNPs with different particle sizes was measured by the growth inhibition curve method, and the antibacterial activity of these nanoparticles against *E. coli* (*ATCC 25922*) and *S. aureus* (*ATCC 25923*) was observed by an optical microscope. *E. coli* (*ATCC 25922*) and *S. aureus* (*ATCC 25923*) were selected as representatives of Gram-negative bacteria and Gram-positive bacteria, respectively.

Firstly, the suspensions of *E. coli* (*ATCC 25922*) and *S. aureus* (*ATCC 25923*) were prepared by shaking culture at 37°C for 8 h in 150 r.p.m. incubator and inoculated in a 96-well plate. The initial density of bacteria was $1 \times 10^7$ CFU ml$^{-1}$, and the final volume was 0.2 ml. Then, different concentrations of 1.8 ml CGA-AuNPs were added as the experimental group, and ultra-pure water was used instead of CGA-AuNPs as the control group. The 96-well plate was placed at 37°C and cultured in 150 r.p.m. incubator for 12 h. The OD600 value was measured every 1 h. The experiment was repeated three times, and three parallel samples were set at a time. ANOVA was used to examine the effects of different concentrations, sizes and strains on bacteriostatic effects.

The bacteria of the experimental group and the control group cultured for 8 h were stained with Gram and their morphology and quantity were observed under an oil microscope.

# 4. Conclusion

In this study, antibacterial CGA-AuNPs ranged from $18.94 \pm 1.81$ to $48.72 \pm 6.47$ nm were successfully synthesized using CGA. CGA-AuNPs with small particle size have a stronger antibacterial effect. And they also have good stability.

Data accessibility. Data available from the Dryad Digital Repository: https://doi.org/10.5061/dryad.6hdr7sqw6 [30].
Authors' contributions. Y.S. made substantial contributions to conception, design, analysis, interpreted data and drafted the manuscript; Y.Y. participated in data analysis and the design of the study; X.B. carried out the statistical analyses and collected field data; S.Z. conceived of the study, designed the study and critically revised the manuscript. All authors gave final approval for publication.

Competing interests. There are no conflicts of interest to declare.

Funding. This work was supported by the Yangzhou University Cooperation Foundation under grant no. 204020813.

Acknowledgements. We are grateful to the Yangzhou University Cooperation Foundation for providing financial support for our research. We are also grateful to Teacher Chen, Dr. Yifang Chen and Dr. Zixia Lin from the Testing Center of Yangzhou University for their help in TEM and HRTEM observation.

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
