## [Reviewer comments · Royal Society Open Science]

Review History

RSOS-191141.R0 (Original submission)

Review form: Reviewer 1

Is the manuscript scientifically sound in its present form?

Yes

Are the interpretations and conclusions justified by the results?

Yes

Is the language acceptable?

Yes

Do you have any ethical concerns with this paper?

No

Have you any concerns about statistical analyses in this paper?

Yes

Recommendation?

Accept with minor revision (please list in comments)

Comments to the Author(s)

The author described a synthesis of gold nanoparticles with different size and evaluated their size-dependent antibacterial performance. This manuscript seems enough good to appeal to the readership of this journal. I suggests its publication, with minor revision.

Two comments:

1. It is suggested that authors need to provide original TEM images of Fig.3.
2. Statistical analysis of Fig. 5 and Fig. 6 should be added.

Review form: Reviewer 2

Is the manuscript scientifically sound in its present form?

No

Are the interpretations and conclusions justified by the results?

No

Is the language acceptable?

No

Do you have any ethical concerns with this paper?

No

Have you any concerns about statistical analyses in this paper?

Yes

Recommendation?

Reject

Comments to the Author(s)

Zhu et al., present their article entitled "Synthesis of antibacterial gold nanoparticles with different particle sizes using chlorogenic acid," where they explored the synthesis of CGA capped nanogold synthesis. The authors have tried to elaborate a convincing article for studying the use of chlorogenic acid as a capping agent able to tune nanoparticle diameter. However, similar approaches had been widely explored in the literature and date from back in 2012 (see for example Langmuir, 2012, 28(21): 8183–8189). In the reasoning of the article, the authors have chosen sodium citrate as a reducing agent, which is a sacrificial component, and also an excellent protecting agent. However, other methodologies have been widely reported in the literature as capable of producing nanoparticles with different sizes, and even geometries see for example Pure and Applied Chemistry, 2011, 83 (4), 913-930. In the details below included are found additional comments:

- i. The article lacks statistical analysis to compare the different size distributions of the nanoparticles prepared alongside with estimations for the sphericity of the produced nanoparticles. 1-way ANOVA analysis is required to compare the different populations and draw sounded conclusions. Note that if deviations of perfectly spherical nanoparticles are found, those should be acknowledged.
- ii. Nanoparticle stability under high ionic strengths is critical when determining whether the observed antimicrobial effect is exerted from the nanoparticle or oxidation side products. Thus, the authors must demonstrate the particles are stable in LB. Initial bacterial density is also missing.
- iii. Strain codes for the bacteria used are not included. Overall, the experimental section is poorly written, which would make it almost impossible to reproduce this study.

iv. The authors have also found CGA alone has some “antimicrobial” properties, which makes the article even more puzzling. The authors need to elaborate further how do they know the observed antimicrobial effect does come from the nanoparticle and not from the protecting agent, which is free in solution.

In summary, the reviewer feels the publication of this work would be premature in the current stage.

Review form: Reviewer 3

Is the manuscript scientifically sound in its present form?

Yes

Are the interpretations and conclusions justified by the results?

Yes

Is the language acceptable?

No

Do you have any ethical concerns with this paper?

No

Have you any concerns about statistical analyses in this paper?

No

Recommendation?

Major revision is needed (please make suggestions in comments)

Comments to the Author(s)

The authors reported the synthesis and antibacterial treatment of Au NPs with the size distribution from 14.4 nm, to 51.6 nm. It is interesting. However, I read the manuscript several times and found it very preliminary with little data. Overall, I think the paper can be accepted for publication in Royal Society Open Science if the authors should clarify the following unclear questions.

- 1) To optimize synthesis condition, is it possible to make Au NPs with the average size less than 10 nm using chlorogenic acid? Figure 3 is less convincing for Royal Society Open Science readers. I strongly suggest the authors make the detailed high-resolution TEM characterization with electron diffraction pattern of Au NPs.
- 2) The antibacterial activity of these CGA-AuNPs should be viewed using optical microscopy.
- 3) The text in figures is too small to clear, and the English in text should be finely polished.

Decision letter (RSOS-191141.R0)

02-Sep-2019

Dear Dr Shen:

Title: Synthesis of antibacterial gold nanoparticles with different particle sizes using chlorogenic acid

Manuscript ID: RSOS-191141

The editor assigned to your manuscript has now received comments from reviewers. We would like you to revise your paper in accordance with the referee and Subject Editor suggestions which can be found below (not including confidential reports to the Editor). Please note this decision does not guarantee eventual acceptance.

Please submit your revised paper before 25-Sep-2019. Please note that the revision deadline will expire at 00.00am on this date. If we do not hear from you within this time then it will be assumed that the paper has been withdrawn. In exceptional circumstances, extensions may be possible if agreed with the Editorial Office in advance. We do not allow multiple rounds of revision so we urge you to make every effort to fully address all of the comments at this stage. If deemed necessary by the Editors, your manuscript will be sent back to one or more of the original reviewers for assessment. If the original reviewers are not available we may invite new reviewers.

Please also include the following statements alongside the other end statements. As we cannot publish your manuscript without these end statements included, if you feel that a given heading is not relevant to your paper, please nevertheless include the heading and explicitly state that it is not relevant to your work.

- Funding statement

Please include a funding section after your main text which lists the source of funding for each author.

RSC Associate Editor:

Comments to the Author:

Significant additional experimental work is required.

RSC Subject Editor:

Comments to the Author:

(There are no comments.)

Reviewers' Comments to Author:

Reviewer: 1

Comments to the Author(s)

The author described a synthesis of gold nanoparticles with different size and evaluated their size-dependent antibacterial performance. This manuscript seems enough good to appeal to the readership of this journal. I suggests its publication, with minor revision.

Two comments:

1. It is suggested that authors need to provide original TEM images of Fig.3.
2. Statistical analysis of Fig. 5 and Fig. 6 should be added.

Reviewer: 2

Comments to the Author(s)

Zhu et al., present their article entitled "Synthesis of antibacterial gold nanoparticles with different particle sizes using chlorogenic acid," where they explored the synthesis of CGA capped nanogold synthesis. The authors have tried to elaborate a convincing article for studying the use of chlorogenic acid as a capping agent able to tune nanoparticle diameter. However, similar approaches had been widely explored in the literature and date from back in 2012 (see for example Langmuir, 2012, 28(21): 8183–8189). In the reasoning of the article, the authors have chosen sodium citrate as a reducing agent, which is a sacrificial component, and also an excellent protecting agent. However, other methodologies have been widely reported in the literature as capable of producing nanoparticles with different sizes, and even geometries see for example Pure and Applied Chemistry, 2011, 83 (4), 913-930. In the details below included are found additional comments:

- i. The article lacks statistical analysis to compare the different size distributions of the nanoparticles prepared alongside with estimations for the sphericity of the produced nanoparticles. 1-way ANOVA analysis is required to compare the different populations and draw sounded conclusions. Note that if deviations of perfectly spherical nanoparticles are found, those should be acknowledged.
- ii. Nanoparticle stability under high ionic strengths is critical when determining whether the observed antimicrobial effect is exerted from the nanoparticle or oxidation side products. Thus, the authors must demonstrate the particles are stable in LB. Initial bacterial density is also missing.
- iii. Strain codes for the bacteria used are not included. Overall, the experimental section is poorly written, which would make it almost impossible to reproduce this study.
- iv. The authors have also found CGA alone has some "antimicrobial" properties, which makes the article even more puzzling. The authors need to elaborate further how do they know the observed antimicrobial effect does come from the nanoparticle and not from the protecting agent, which is free in solution.

In summary, the reviewer feels the publication of this work would be premature in the current stage.

Reviewer: 3

Comments to the Author(s)

The authors reported the synthesis and antibacterial treatment of Au NPs with the size distribution from 14.4 nm, to 51.6 nm. It is interesting. However, I read the manuscript several times and found it very preliminary with little data. Overall, I think the paper can be accepted for publication in Royal Society Open Science if the authors should clarify the following unclear questions.

- 1) To optimize synthesis condition, is it possible to make Au NPs with the average size less than 10 nm using chlorogenic acid? Figure 3 is less convincing for Royal Society Open Science readers. I strongly suggest the authors make the detailed high-resolution TEM characterization with electron diffraction pattern of Au NPs.
- 2) The antibacterial activity of these CGA-AuNPs should be viewed using optical microscopy.
- 3) The text in figures is too small to clear, and the English in text should be finely polished.

Author's Response to Decision Letter for (RSOS-191141.R0)

See Appendix A.

RSOS-191141.R1 (Revision)

Review form: Reviewer 2

Is the manuscript scientifically sound in its present form?

No

Are the interpretations and conclusions justified by the results?

No

Is the language acceptable?

No

Do you have any ethical concerns with this paper?

No

Have you any concerns about statistical analyses in this paper?

Yes

Recommendation?

Reject

Comments to the Author(s)

Zhu et al., present a revised version of their article entitled "Synthesis of antibacterial gold nanoparticles with different particle sizes using chlorogenic acid," where they explored the synthesis of CGA capped nanogold synthesis. The authors have added new experimental data

and tried to amend the Dr comments. However, they have failed at addressing the following points:

- i. Statistical analysis is still lacking proper elaboration. T-tests do not work when analysing multiple samples. The authors should, as originally recommended, use ANOVA.
- ii. Data showed in Figure 3 need to be compared using statistical analysis. A direct comparison between the histograms is not accurate and rather misleading. The number of nanoparticles measured per group is also missing.
- iii. From the data showed in Figs. 6F, 7F, 7G, and 7H it seems that the nanoparticulated material has equal, or at least comparable, antimicrobial potency than CGA alone. The authors need to address why there is a need for having nanoparticles if the reducing agent CGA does the same.
- iv. Data included in Figure 8 need of the CGA treated cells for drawing any valid conclusions.

Review form: Reviewer 3

Is the manuscript scientifically sound in its present form?

Yes

Are the interpretations and conclusions justified by the results?

Yes

Is the language acceptable?

Yes

Do you have any ethical concerns with this paper?

No

Have you any concerns about statistical analyses in this paper?

No

Recommendation?

Accept as is

Comments to the Author(s)

I failed to find a response letter to reviewers. Anyway, the quality of the revised manuscript has partially improved; I think it may be now acceptable for publication in in Royal Society Open Science, but my overall rating is "fair".

Decision letter (RSOS-191141.R1)

19-Nov-2019

Dear Professor Zhu:

Title: Synthesis of antibacterial gold nanoparticles with different particle size using chlorogenic acid

Manuscript ID: RSOS-191141.R1

The editor assigned to your paper has now received comments from reviewers. We would like you to revise your paper in accordance with the referee and Subject Editor suggestions which can be found below (not including confidential reports to the Editor). Please note this decision does not guarantee eventual acceptance.

Please submit a copy of your revised paper before 12-Dec-2019. Please note that the revision deadline will expire at 00.00am on this date. If we do not hear from you within this time then it will be assumed that the paper has been withdrawn. In exceptional circumstances, extensions may be possible if agreed with the Editorial Office in advance. We do not allow multiple rounds of revision so we urge you to make every effort to fully address all of the comments at this stage. If deemed necessary by the Editors, your manuscript will be sent back to one or more of the original reviewers for assessment. If the original reviewers are not available we may invite new reviewers.

RSC Associate Editor:
Comments to the Author:
Reviewer 2 requires further validation of experimental work before publication.

RSC Subject Editor:
Comments to the Author:
(There are no comments.)

Reviewers' Comments to Author:

Reviewer: 3

Comments to the Author(s)

I failed to find a response letter to reviewers. Anyway, the quality of the revised manuscript has partially improved; I think it may be now acceptable for publication in Royal Society Open Science, but my overall rating is "fair".

Reviewer: 2

Comments to the Author(s)

Zhu et al., present a revised version of their article entitled "Synthesis of antibacterial gold nanoparticles with different particle sizes using chlorogenic acid," where they explored the synthesis of CGA capped nanogold synthesis. The authors have added new experimental data and tried to amend the Dr comments. However, they have failed at addressing the following points:

- i. Statistical analysis is still lacking proper elaboration. T-tests do not work when analysing multiple samples. The authors should, as originally recommended, use ANOVA.
- ii. Data showed in Figure 3 need to be compared using statistical analysis. A direct comparison between the histograms is not accurate and rather misleading. The number of nanoparticles measured per group is also missing.
- iii. From the data showed in Figs. 6F, 7F, 7G, and 7H it seems that the nanoparticulated material has equal, or at least comparable, antimicrobial potency than CGA alone. The authors need to address why there is a need for having nanoparticles if the reducing agent CGA does the same.
- iv. Data included in Figure 8 need of the CGA treated cells for drawing any valid conclusions.

Author's Response to Decision Letter for (RSOS-191141.R1)

See Appendix B.

RSOS-191141.R2 (Revision)

Review form: Reviewer 3

Is the manuscript scientifically sound in its present form?

Yes

Are the interpretations and conclusions justified by the results?

Yes

Is the language acceptable?

Yes

Do you have any ethical concerns with this paper?

No

Have you any concerns about statistical analyses in this paper?

No

Recommendation?

Accept as is

Comments to the Author(s)

The quality of the revised manuscript R2 has improved; I think it may be now acceptable for publication in Royal Society Open Science.

Decision letter (RSOS-191141.R2)

06-Jan-2020

Dear Professor Zhu:

Title: Synthesis of antibacterial gold nanoparticles with different particle sizes using chlorogenic acid

Manuscript ID: RSOS-191141.R2

It is a pleasure to accept your manuscript in its current form for publication in Royal Society Open Science. The chemistry content of Royal Society Open Science is published in collaboration with the Royal Society of Chemistry.

RSC Associate Editor:
Comments to the Author:
(There are no comments.)

RSC Subject Editor:
Comments to the Author:
(There are no comments.)

Reviewer(s)' Comments to Author:

Reviewer: 3

Comments to the Author(s)

The quality of the revised manuscript R2 has improved; I think it may be now acceptable for publication in Royal Society Open Science.

Appendix A

Dear Editors and reviewers

I am the corresponding author of the manuscript (**ID: RSOS-191141**) entitled “**Synthesis of antibacterial gold nanoparticles with different particle size using chlorogenic acid**”. First of all, on behalf of all the authors of this manuscript, I would like to thank the editors of Royal Society Open Science, RSC editors and reviewers for processing our manuscript and the constructive reviews on our manuscript. Your constructive comments are very helpful for us to improve our manuscript. We have read all the comments carefully. Based on your comments, we have complemented some data and revised the manuscript. The modified text is marked with a red mark in the revision mode of Microsoft Word. Now, we submit the revised manuscript to **Royal Society Open Science**.

Our responses to each of the reviewers’ comments have been appended to this letter.

Thank you again for your reconsidering our manuscript.

Sincerely yours

Sujuan Zhu

Associate Professor of Biochemistry and Molecular Biology
College of Bioscience and Biotechnology
Yangzhou University
Yangzhou City, Jiangsu, China 225009

Responses to each of the reviewers' comments

Responses to reviewer 1's comments

Ref. to point 1: We have replaced the TEM image with the original TEM image. Please see Fig. 3.

Ref. to point 2: The statistical analysis of Fig. 5 and Fig. 6 in the first draft has been supplemented in the revised manuscript. Please see Fig. 6. and Fig. 7.

Responses to reviewer 2's comments

We have carefully read the two articles of Langmuir, 2012, 28(21): 8183–8189 and Pure and Applied Chemistry, 2011, 83 (4), 913-930, and the content of the report is very excellent research work on photochemically synthesized nanoparticles, which we have cited. Our synthetic methods are characterized by: 1) chemical synthesis, simple, fast and low cost; 2) introduction of natural compound CGA as a substance to regulate particle size. We believe that the controllable synthesis of nanoparticles is of great value for its practical application. However, the technology for controlling the size and shape of nanoparticles is not yet fully mature, especially in the regulation of synthesis using natural compound, so continuous research in this area is needed. Related descriptions have been modified in the abstract and introduction section.

Ref. to point 1: In order to better compare the particle size distribution of different particle size CGA-AuNPs, we unified the statistical quantities of CGA-AuNPs of different particle sizes into 100 individual nanoparticles, and re-drawn the particle size distribution histogram. Please see the Fig. 3. We have failed to assess the sphericity of CGA-AuNPs because of certain difficulties in technical equipment. The synthesized CGA-AuNPs is not a perfect spherical shape, so there is some deviation, and we have acknowledged in the revised manuscript. But the results of TEM and UV-vis spectrum can be visually verified that the particles become larger as the CGA concentration increases.

Ref. to point 2: We have completed experimental data to show that CGA-AuNPs were stable in LB. Please see Fig. 5 B. The initial density of bacteria has been supplemented in the revised manuscript.

Ref. to point 3: We have added the bacterial strain code to the text, and the

description of the bacteriostatic experiment has been revised. Please see the revised manuscript.

Ref. to point 4: CGA is a natural compound with antibacterial effect. We used it as a control group and different particle size CGA-AuNPs as the experimental group, so we determined their effects on the bacterial growth curve. We have added their statistical analysis to show that the antibacterial effect of CGA-AuNPs may be derived from both CGA and nanoparticles. Please see Fig. 6 and Fig. 7.

Responses to reviewer 3's comments

Ref. to point 1: We have optimized the synthesis conditions to try to synthesize CGA-AuNPs with an average particle size of less than 10 nm using CGA, but the results showed that it was not successful. And we have planned it for the next work that needs to be explored. High-resolution transmission electron microscopy and electron diffraction characterization of AuNPs have been added. Please see Fig. 4.

Ref. to point 2: We have observed the antibacterial activity of CGA-AuNPs by optical microscopy. Please see Fig. 8.

Ref. to point 3: We have already enlarged the text in the pictures, and the English in the revised version has been carefully checked and modified. Please see revised manuscript.

Appendix B

Responses to reviewers' comments

Responses to reviewer 2's comments

Ref. to point 1: The bacteriostatic effect of CGA-AuNPs on two kinds of bacteria has been statistically analyzed by using ANOVA instead of T-tests. Relevant results please see the part of *Bacteriostatic effect of CGA-AuNPs* in the revised manuscript.

Ref. to point 2: Data showed in Fig. 3 has been compared using One-way ANOVA. The number of nanoparticles measured per group has been added in the text. Relevant results please see the part of *TEM analysis of CGA-AuNPs* and Fig.3 F in the revised manuscript.

Ref. to point 3: The experimental results showed that the antibacterial effect of CGA-AuNPs is related to particle size and concentration, and the antibacterial effect of CGA-AuNPs ($18.94 \pm 1.81\text{nm}$) is basically the same as CGA that under the same conditions. However, compared with CGA, CGA-AuNPs are not redundant. They are two different individuals with different properties. CGA-AuNPs are nano materials, which have a lot of potential that CGA does not have. It has a lot of potential to be developed, such as further modifying CGA-AuNPs to achieve its target binding function in vivo, so that it can achieve bacteriostasis in the place where we need bacteriostasis. At this time, the advantages of good biocompatibility and low cytotoxicity of nano gold will be well demonstrated.

The content of this manuscript is the first result of our team's research on CGA-AuNPs, and our ultimate goal is to achieve practical application. Therefore, we hope to publish it in the *Journal of royal society open science*, so that CGA-AuNPs can be seen by more researchers who are interested in it, and then jointly explore many functions of CGA-AuNPs that need to be unlocked until practical application. Relevant instructions please see the part of *Bacteriostatic effect of CGA-AuNPs* in the revised manuscript.

Ref. to point 4: The content of CGA treated bacteria have been added to the revised manuscript. Please see the part of *Bacteriostatic effect of CGA-AuNPs* and Fig. 8 B&G.